Natural Hazards of Sciences

# Impacts of extreme weather events on transport infrastructure in Norway

Regula Frauenfelder<sup>1</sup>, Anders Solheim<sup>1</sup>, Ketil Isaksen<sup>2</sup>, Bård Romstad<sup>3a</sup>, Anita V. Dyrrdal<sup>2</sup>,

5 Kristine H. H. Ekseth<sup>1</sup>, Alf Harbitz<sup>4</sup>, Carl B. Harbitz<sup>1</sup>, Jan Erik Haugen<sup>2</sup>, Hans Olav Hygen<sup>2</sup>, Hilde Haakenstad<sup>2</sup>, Christian Jaedicke<sup>1</sup>, Árni Jónsson<sup>1</sup>, Ronny Klæboe<sup>5</sup>, Johanna Ludvigsen<sup>5</sup>, Nele M. Meyer<sup>1</sup>, Trude Rauken<sup>3b</sup>, Reidun G. Skaland<sup>2</sup>, Kjetil Sverdrup-Thygeson<sup>1</sup>, Asbjørn Aaheim<sup>3</sup>, Heidi Bjordal<sup>6</sup>, Per-Anton Fevang<sup>7</sup>

<sup>1</sup>Norwegian Geotechnical Institute, Oslo, Norway

<sup>2</sup> The Norwegian Meteorological Institute, Oslo, Norway
 <sup>3</sup> CICERO Centre for Climate and Environmental Research, Oslo, Norway
 <sup>4</sup> Institute of Marine Research, Bergen, Norway
 <sup>5</sup> Institute of Transport Economics, Oslo, Norway
 <sup>6</sup> Norwegian Public Roads Administration, Oslo, Norway
 <sup>7</sup> Bane NOR, Oslo, Norway

<sup>a</sup> now at: Amedia, Oslo, Norway <sup>b</sup> now at: Municipality of Oslo, Oslo, Norway

Correspondence to: R. Frauenfelder (rf@ngi.no)

- Abstract. This paper presents selected results of the interdisciplinary research project "Impacts of extreme weather events on infrastructure in Norway (InfraRisk)"carried out between 2010 to 2013, as part of the program NORKLIMA (2004-2013) of the Research Council of Norway (RCN). The project has systematized large amounts of existing data and generated new results that are important for our handling of risks associated with future extreme weather and natural hazards threatening the transport infrastructure in Norway. The results of the InfaRisk project range widely, from the establishment of trends in key weather
- elements to studies of human response to threats from extreme weather. The analyses of weather elements have provided a clearer understanding of the trends in the development of extreme weather. The studies are based on both historical data and available future scenarios (projections) from climate models. Compared to previous studies, we calculated changes in climate variables that are particularly important in relation to nature hazards. Overall, the analyses document an increase in frequency as well as intensity of both precipitation and wind. Results of projections show that the observed changes will continue
- throughout this century. We could also identify large regional differences, with some areas experiencing, e.g., a reduction in the intensity of heavy rainfall events. However, most of the country will experience the opposite, i.e., both increased intensity and increased frequency of heavy precipitation. Our analyses show that at least 27 per cent of Norwegian roads and 31 per cent of railroads are exposed to rock fall and snow avalanches hazards. The project has also assessed relationships between different parameters that can affect the likelihood of debris flows. Variables such as terrain slope and size of watercourses are
- important, while local climate, which varies widely in Norway, determines threshold values for rainfall that can trigger debris flows.

## **1** Introduction

# 1.1 Background

A number of studies, the most prominent being the IPCC reports (e.g., IPCC, 2013) show that global warming leads to increase

- in both frequency and intensity of several types of extreme weather, in particular when it comes to precipitation. With the latest results on expected future increase in air temperature and precipitation changes reported by the Intergovernmental Panel on Climate Change (IPCC, 2013), the climate robustness of important infrastructure is of raising concern in Norway, as well as in the rest of Europe. Economic consequences of natural disasters have increased considerably since 1950 (IPCC, 2012; Munich RE, 2016). This not only due to demographic changes such as population growth, urbanization, increased economic
- activity and transport movements with a concentration of valuable assets, but also related to regional increase in the intensity and frequency of strong precipitation events. A report by Hanssen-Bauer et al. (2015) shows a clear increase in both average temperature and precipitation for Norway during the last century. Annual precipitation since the year 1900 has risen by 27 percent in some areas (Hanssen-Bauer et al., 2015; Figure 1a). Moreover, previous studies show an increase in the annual number of rainfall days in several locations in Norway (Alfnes and Førland, 2006). Similar trends are found for air temperature
- (Figure 1b), but with larger regional variations (Hansen-Bauer et al., 2015). Benestad (2013) showed a statistically significant positive relationship between the increase in the global mean temperature and the global occurrence of intense precipitation. Such changes are also observable in Norway, where the increased frequency of intense precipitation (Dyrrdal et al., 2012) has led to frequent rainfall-induced flooding and landslide events during the last decades (Wilson et al., 2010; Vormoor et al., 2016).
- Extreme weather events greatly affect the transport infrastructure, with numerous closures of roads and rail networks, in addition to damage and repair costs. Disruptions and closures of railways and roads lead to delay or failure in delivery of goods, and consequences can propagate up- and downstream through the supply and demand networks. A robust infrastructure is a competitive advantage, and businesses and transport activities that are vulnerable to disruptions and delay run the risk of being replaced by competitors with higher resilience. In Norway, where the railroad network is sparse, the economic
- consequences of extreme weather events during the past decades, has led to a shift from transport of goods by rail to road haulage. Similar trends are seen in other parts of Europe as well, particularly after the extreme winters with heavy snowfall and long cold periods in 2010 and 2011, respectively (Ludvigsen and Klæboe, 2014). Extreme weather events have negative effects not only on the road quality (with the increased wear of roads), but more importantly, on disruption of services. The latter may lead to reduction in the competitiveness of environmentally friendly rail transport, thus contributing to increased
- CO<sup>2</sup> emissions from goods supply. In order to meet the Norwegian government's goals of a climate neutral Norway by 2030, significant reductions of emissions from all sources are needed. Moving cargo and passengers from roads to railways is one of the proposed means to reach this goal. However, the railway infrastructure will need significant investments to handle increased traffic, not the least to improve transport reliability under present and future challenging climate conditions. Much of the Norwegian transport infrastructure is more than 50 years old and therefore not adequately designed even for
- present climate conditions. In particular, this is the case for Norwegian railways. Studies both in Norway and in Switzerland (Rauken, 2009; Bründl et al., 2009) show that upgrading of infrastructure or building of mitigation measures against natural hazards rarely are done until after an event has caused damage. Although the installment of measures is expensive, the costs of not taking actions will always be greater after a damage incident, when the cost of repair and restoration must be added, in addition to the other, long-term negative consequences mentioned above.

To assess the problems and challenges posed to the Norwegian transport infrastructure from present and future extreme weather events, the project "Impacts of extreme weather events on infrastructure in Norway (INFRARISK)" was carried out between 2009 and 2013, under the Research Council of Norway's program 'NORKLIMA'. Important background material for INFRARISK came from former national projects, such as those reported by Jaedicke et al. (2008), Petkovic (2013), Hanssen-

5 Bauer et al. (2009), and a governmental interagency project (the 'NIFS' project), which was partly carried out during the same time period as the INFRARISK project (Dolva et al., 2012; Aunaas et al., 2016).

# 1.2 Scope and objectives

In order to design the Norwegian transport infrastructure to tackle present climate, as well as future climate conditions, one must establish systematic knowledge about: (a) the extreme weather events with the most serious impact on transport

- infrastructure, (b) the climate variables that lead to these events, and (c) how the distribution, intensity and frequency of such events will change. The links between extreme weather events and hazards, such as landslides and flooding, and how these combined hazards may affect the transport infrastructure also need to be assessed. The main objectives of the INFRARISK project (NGI, 2013a) therefore were fivefold: (1) To better understand the links between climate change and the frequency, intensity and distribution of extreme weather events in Norway. (2) To establish a knowledge base of the effect of extreme
- weather events on the Norwegian transport infrastructure. (3) To quantify the vulnerability and the social economic importance of the infrastructure. (4) To determine which mitigation measures are the most efficient as means of climate adaptation for the transport infrastructure. (5) To develop an improved methodology for risk assessment of the transport infrastructure in relation to extreme weather events. The present paper gives a summary of the InfraRisk project and presents a selection of its main results.

#### 20 2 Methods

The resilience of Norwegian transport infrastructure relative to their increased exposure to extreme events was approached through a broad set of analyses of weather elements, natural hazards, and their effects and impacts. An overview of the methods that have been used is given below; some more details are given in the individual results chapters. For thorough descriptions of the applied methodology, we refer to – where applicable – already published project results.

## 2.1 Changes in frequency and intensity of extreme weather events

The most frequent natural hazard in Norway are snow avalanches, debris slides/flows, rock fall, and flooding (e.g., Jaedicke et al., 2008; Dolva et al., 2012), most of which are triggered and/or aggravated by extreme weather conditions. The study has analyzed both the historical and the potential future development of frequency and intensity of the climate variables that may trigger these natural hazards.

trigger these natural hazards.

# 2.1.1 Historical trends

#### Climate variables used for the historical analyses are listed in Tables

Table 1. These variables are derived from datasets of daily temperature and precipitation measurements interpolated on a 1 x 1 km<sup>2</sup> grid, available for entire Norway from 1957 to present (cf. Tveito et al., 2005; Jansson et al., 2007; Mohr, 2009).

Temperature and precipitation values were used as input into a hydrological model that simulates the various snow parameters (Engeset et al., 2004). Uncertainties in the dataset are primarily caused by the complex topography of Norway and a variable

density of observation points for the interpolated dataset. Typically, the precipitation vales are overestimated in higher elevations and somewhat underestimated in lower elevations. Further uncertainty is introduced by the use of a relatively simple snow model (Saloranta, 2012). Despite some deficiencies, these datasets were the best available ones on a national basis at the time of the INFRARISK project.

- Trend analyses for selected variables were performed using the Mann-Kendall trend test for various periods. Statistical significance was assessed by carrying out field significance tests for 19 regions and, subsequently, calculating mean regional changes. For the trend analysis of short-term precipitation (1–24 hours) we had access to a historical hind-cast archive (NORA 10) with a spatial resolution of ca. 11 km, produced by the Norwegian Meteorological Institute (Reistad et al., 2007; Reistad et al., 2011). The archive was produced by combining the HIRLAM weather forecast model (version 6.4.2, Undén et al., 2002)
- and the ERA40 reanalyzes (Uppala et al., 2005) for the period September 1957 through August 2002. For the remaining period, 2002 until present, the boundary values were taken from the operational weather forecasting model at ECMWF (European Centre for Medium-Range Weather Forecasts). Validating the NORA 10 archive against 70 Norwegian weather stations showed the archive to be the best available replacement for (lacking) observation data in Norway, particularly for precipitation values over 5-10 mm/day. We have analyzed the period 1961–2010, and compared the two 30-year periods 1961–1990 and
- 1981–2010. The applied methodology is presented detail in Dyrrdal et al. (2012).

## 2.1.2 Future trends

Seven existing scenarios from General Circulation Models (GCMs) were used in our analyses, which were downscaled with dynamical methods to obtain improved spatial resolution with three different RCMs. These RCMs where the Norwegian HIRHAM model (Haugen and Haakenstad, 2006, Engen-Skaugen et al., 2007), the Danish HIRHAM model (Jungclaus et al.,

2006; Roeckner et al., 2006) and the Swedish RCA3 model (Déqué et al., 1994; Bleck et al., 1992). Three of the model runs were performed within the ENSEMBLES project (http://ensemblesrt3.dmi.dk). The downscaled scenarios were statistically adjusted, using topographic information and observations from a grid with 1 x 1 km resolution (Engen-Skaugen, 2007). An overview of the used scenarios is given in NGI (2013a).

#### 25 2.2 Assessing the exposure of the transport infrastructure to extreme weather events

The exposure of the Norwegian transport infrastructure to selected hazard types, such as snow avalanches, rock fall and rock slides, was estimated through GIS-analyses. Hereby, the national road and railroad network was combined with existing susceptibility maps for snow avalanches and rockslides (NGI, 1987). Thus, the percentage of the transport network located within the respective hazard zones, and therewith the potential exposure to these two hazard types, was calculated.

- 30 The map series used was originally prepared on governmental mandates and does not cover the entire country, yet, it covers the most slide-prone regions of the country. These susceptibility maps are based on field mapping in combination with numerical modelling. Potential release areas in these maps were identified using slope-angle maps to find terrain where snow avalanches or rockslides can be triggered. Run-out areas were calculated using a statistical run-out model developed by Bakkehøi et al. (1983). The regions flagged as hazardous by the models were then inspected by field-visits and rated by experts,
- 35 before the final delineation of the hazard zones was defined. To a certain extent, the maps also account for debris flow hazards, with the borders of the hazard zones generally being drawn more conservative along stream channels than indicated by the models. This original susceptibility map series was favored over a newer map series, which does cover the enitre country (Høst et al., 2013), but does not involve field surveys. The original maps used here, are far more accurate and less conservative than the latter, purely automatically generated maps.

#### 2.3 Precipitation thresholds for debris flow initiation

The study of precipitation thresholds for debris flow initiation was based on empirical data from 429 documented debris flow events in an area of western Norway. The data set was split in a "training set" and a test set", and areas along national roads and railroads received most focus. For the assessment of trigger conditions, intensity and duration of rainfall events were

- 5 related to the spatio-temporal distribution of debris flows. With the help of empirically derived intensity-duration (ID) relationships, one is able to indicate critical conditions for debris flow initiation. ID thresholds are based on the assumption that a lower limit of rainfall intensity over a given duration exists, which exceedance triggers enhanced debris flow activity (Wieczorek and Glade, 2005). The combination of rainfall intensity and rainfall duration in a threshold indicating critical conditions for debris flow initiation relates to the role of antecedent moisture conditions for debris flow initiation (Wilson and
- 10 Jayko, 1997). Power laws of the form  $I = a^*D^b$  are the most common forms for ID threshold curves described in the literature (Guzzetti et al., 2007; Brunetti et al., 2010) and also used in our study. With sufficient data available, the threshold curve separating debris flow triggering rainfall events from events which did not result in debris flow initiation, can be established. Annual precipitation in Norway varies between >3000mm in the fjord areas of the west coast, to 

Because the risk is calculated for each individual object, the spatial distribution of risk can also be presented in a map. The focus in this project has been on rapid mass wasting and transport infrastructure objects, but the model can also be applied to other types of hazards and objects. The source code of the model is freely accessible on: https://github.com/gisminister/infrarisk (explanations given in Norwegian).

5

## **3 Results**

#### 3.1 Frequency and intensity of extreme weather events in Norway

The following paragraphs present a short summary of results from Dyrrdal et al. (2012), a study which was part of the InfraRisk-project.

## 10 **3.1.1 Precipitation over durations** $\geq$ **1 day**

Annual maximum 1-day and 10-day precipitation have increased in Norway since 1957 (*Fig. 1 and Fig. 2*). The largest increase is seen in areas with already high precipitation in the west and southwest. Patterns are more apparent for longer-duration precipitation and in some regions of the southwest (Fig. 2), where average regional changes are as large as ~90% for 10-day precipitation (Dyrrdal et al., 2012). The trends are confirmed by observational time series from stations in these regions. Some

15 areas, however, also show decreasing trends, but some of this variability may be due to inconsistencies in the station network. The peak over threshold analyses for daily precipitation exceeding 10mm (*Fig. 3*) show a more significant increase in large parts of the country, indicating a larger increase in moderate to strong precipitation events. The values are between 10 and 30% for most of the country over the last 50 years. The relatively low threshold of 10 mm is necessary to obtain a sufficient number of events for trend analysis.

#### 20 3.1.2 Snow and "near-zero events"

Trends in annual maximum snow depth show a general increase in high elevated and inland areas, where winters are usually cold despite an increase in temperature, whereas we see a decrease in low-lands and coastal areas. The trends vary between time periods, with the most significant decrease seen during the last 30-years analyzed (*Fig.4*). These trends correspond well with the observed temperature increase over the last decades, leading to more winter precipitation as rain.

25 Near-zero events (daily mean temperature between -1.5 and 1.0°C), indicative of frequent freeze-thaw events, lead to wear on roads and may trigger rock fall and rockslides. Since the late 1950's, most of Norway shows an increase in the frequency of these conditions, except for the milder coastal areas. This is in good accordance with the generally positive temperature trend.

#### 3.1.3 Short-term precipitation trends

Precipitation values for durations less than one hour are available only for single stations. For periods of 1–24 hours, we have used the NORA 10 archive (Reistad et al. 2007; Reistad et al. 2011). The trends vary between regions in Norway, with the greatest increase in short-term precipitation events in west and southwest Norway, with a weaker increase in other regions, and a small decrease in parts of mid Norway (*Fig 5*). The region of Rogaland county, in southwest Norway, has an increase in the intensity of 1-hour precipitation of 21% between 1961 and 2010.

### 3.1.4 Future extreme precipitation in Norway

The scenario analyses for this century shows that the historical trends continue and strengthen. Annual maximum 1-day precipitation increases for most of Norway, with the highest increase in western Norway and parts of northern Norway (*Fig. 6a*;). Some areas of mid Norway show a small decrease. Even more significant differences are shown by comparing

precipitation over threshold values, such as number of days with more than 10 mm precipitation, between the reference period 1961–1990 and the last 30 years of the present century (*Fig. 6b*). Due to a general increase in temperature, the number of near-zero events, with freeze-thaw cycles, will decrease through the 21'st century.
 In summary, the downscaled regional trends for the rest of the 21'st century seem to continue the trends seen from the historical

data analyses. The precipitation will increase over most of the country on all timescales, with a particular increase in short

term moderate to strong precipitation events, which will occur over the whole country. Areas which already receive most precipitation will see the highest increase.

## 3.2 Exposure of the Norwegian transport infrastructure

## 3.2.1 Exposure to avalanches and rockfall

- A GIS analysis of susceptibility maps for snow avalanches and rockslides, combined with maps of national roads and the public railroad system shows that within the mapped areas, 31% of the railroad system and 27% of the roads are exposed to hazard from snow avalanches and/or rockslides (Fig. 7). These are minimum numbers, since the maps do not cover the entire country. However, the maps cover at least 70% of the most slide prone areas of Norway, so linear extrapolation of the values using the area of the country has little value. Stating that more than 30% of the roads and railroads in Norway are exposed is
- therefore not an exaggeration. This represents significant values and a great challenge when considering adaptation. As mitigation of all exposed transport infrastructure against all natural hazards is impossible, or at least impractical, thorough cost-benefit analyses must form the base for the planning of mitigation measures.

# 3.2.2 Debris flow threshold values.

Precipitation thresholds are important parameters for the linking of landslide threat with climate. Debris flows are a landslide type which frequently disrupt transport routes in Norway. They typically occur in steep valley sides with streams and ravines, cut into a glacial till cover of variable thickness. These events cause significant damage to roads and railroads each year, and is a reason for frequent closures with delays as a result. Based on empirical data, the InfraRisk project established a model for estimating threshold values for debris flow initiation. Establishment of threshold values can prevent damage to roads and railroads as well as other infrastructure, in that it can trigger different actions when thresholds are exceeded. The methods and results of these studies are described in detail by Mayer et al. (2012).

As expected, the threshold values vary greatly throughout the country (*Fig. 8*). The minimum, average and maximum thresholds vary in the range of 6-63 mm/day, 7-131 mm/day, and 12-250 mm/day, respectively, depending on location and the duration of the water supply event (snowmelt and/or rainfall) (Meyer et al., 2012). The study shows regional differences in

35 susceptibility for debris flows with the highest numbers of threshold crossings in west, central, and northern Norway. The orientation of the most susceptible slopes vary geographically, with the minimum and average threshold values being crossed most frequently in slopes facing west, whereas the maximum threshold values seem to be crossed most frequently in slopes facing ast in central Norway and in the north.

Studies of terrain dependency in a selected part of western Norway showed that the most susceptible areas are those with a gradient between  $22^{\circ}$  and  $61^{\circ}$ , and with an areal extent of 0.2 to 2 km<sup>2</sup>. The terrain was further split in four susceptibility classes; very low, low, intermediate and high, covering 79.5%, 14%, 6.1%, and 0.4% of the study area, respectively.

5 Results from the threshold study can be applied for more differentiated early warning of debris flow hazard. Susceptibility based on terrain parameters can be calculated for road and railroad stretches, and the three threshold levels can be used to identify warning levels, triggering different type of measures, such as inspection of drainage, reduced speed for trains, or in worst case, closure of defined railroad or road stretches. More work on validation of the model is needed before it can be applied as described above.

# 10 **3.3.3 Wind exposure.**

Strong winds can pose a hazard to the transport infrastructure in itself, but wind also have side effects like snowdrift, which is a challenge to the infrastructure, particularly in mountain regions. The InfraRisk project produced exposure maps for strong winds to the transport routes, and estimated changes in the wind fields since 1961. The observational data for wind are considerably sparser than for precipitation, and the models are scarcely validated. In particular, the values for the mountain

regions may be underestimated. The trends found (Fig. 9) are however, considered significant and show an increase in the annual maximum wind strengths of approximately 7 to 8 % for most of the country.

## 3.4 Risk assessment and cost estimates

Although physical mitigation measures may reduce economic loss in case of an incident, they always have a high initial cost. Furthermore, in a country like Norway, with large parts of the country being slide-prone, complete protection against natural

hazards such as avalanches, rockslides and landslides is not realistic. Therefore, prioritizing construction of measures must be based on cost-benefit analyses. Main elements of this, treated in the InfraRisk project, include the assessment of probability for hit by an event, the estimation of different cost elements for infrastructure hit by an avalanche or landslide, and the establishment of a risk model to be applied for the different types of infrastructure.

## 3.4.1 Probability of hit

- The 'Norwegian Planning and Building Act' defines the level of acceptable hazard for different types of built infrastructure regarding natural hazards in Norway. The levels are defined by estimated return periods and expressed as nominal annual probability. As an example, the location of a family dwelling must have a nominal annual probability for being hit by an event of less than 1/1000. As a consequence, hazard mapping in Norway follows the regulations of the Building and Planning Act, and boundaries are defined for annual probabilities of 1/100, 1/1000, and 1/5000, respectively. However, little is done
- regarding a more differentiated estimation of probability for hit. For roads, the accepted annual nominal probability is a function of the traffic volume, defined as 'yearly day traffic' (NPRA, 2014). This implies that robust estimates of probability for hit by avalanches or landslides at any given location would be a very important tool in prioritizing construction of mitigation measures. The most commonly used method for estimating the hazard levels for snow avalanches in Norway is the "α β model" (Bakkehøi et al., 1983), which is a topographical statistical model based on a number of registered avalanches in
- Norway. A similar model developed for landslides has a more uncertain statistical base than for avalanches. A revised  $\alpha \beta$  model developed by Harbitz et al. (2001) was used in the InfraRisk project to better estimate the probable run-out at any given point along an avalanche path. The model requires a run-out less than maximum, and that the recurrence intervals are known in at least one point along the avalanche path. The revised model has been tested in known avalanche paths along the railroad Raumabanen in western Norway. The product of the probability for run-out to a given point along an avalanche path and the

probability for release, gives the probability for hit at a given point along the path. The results indicate that the recurrence intervals found by use of the  $\alpha - \beta$  model along the tested avalanche paths may be too long by a factor of 5-10. This would of course have a large impact on the risk estimates for road or railroad crossing the avalanche path. However, more testing and refining of the model must be performed before it can be applied with a sufficient certainty. The method is described in more detail by NGI (2013b).

3.4.2 Cost analysis

The risk model described above (section 2.4) was first applied in a case study for the town Otta in central southeast Norway. Otta is exposed to snow avalanches, rockslides, debris flows and flooding, and comprises both railroad network sections, and

10 different levels of the road network in Norway. Hazard zoning has been performed for all the named natural hazards in Otta (Fig. 10), which gives a good backdrop for the application of the risk assessment model. In the calculations below, only the road network is included.

The model calculates 2.2 events per year in the Otta case study area, leading to 0.3 annual closure events. Annual costs are estimated to 330 000 NOK (≈ 35 000 Euros), which is relatively modest. Debris flows are most frequent and are the cause for most of these closures. The model results are linked to the map objects, thus the elements that contribute most to the closure frequency are easily identified in the maps. This can be an important tool for local planners. In the Otta case, model results showed that mitigation at the five most exposed points could be expected to reduce the number of closures by 40%.

20 To obtain an overview for the whole country, the model was applied on all major roads (Europe-roads, state roads and county roads) in Norway. Where national hazard maps were missing, data on actual events from the National Public Road Authorities – NPRA - (Bjordal and Helle, 2011) for the period 2000 to 2010 were used to estimate return periods.

The model predicted a total of 1050 events hitting roads annually, resulting in 223 closures with a total closure time of 3000 hours per. year. Based on this, total costs were estimated to roughly 11 million Euros per year (90 million NOK, 2005-prices), of which the road closures comprise about 70%. Rock fall events have the highest frequency, whereas snow avalanches lead to the highest number of closures. The numbers are unevenly distributed throughout the country (Fig. 11), reflecting topographic and climatic variability in Norway.

By comparison Bråthen et al. (2008) calculated the annual closure rate for the Norwegian main road network to 136 events, the annual closure period to 2233 hours and annual closing costs amounting to around 7 million Euros per year. Our analyses include events over a longer period and while Bråthen et al. (2008) based their calculations on the average value for both amount of traffic, closing time and detour length, we have estimated the costs based on the registered actual traffic volume at each hazard point.

The number of closures and the resulting economic losses are most likely underestimated, since only closures caused by actual avalanches, rock falls and landslides are included in the analysis. By analyzing a log of NPRAs live traffic message service for the years 2013-2015 we found reports of a total of 1370 closures due to such events (some 450 per year). These events caused a total closure time of 6300 hours (2100 hours per year). Additionally there were reports of 875 closures due to

40 landslide/rock-fall/avalanche *danger* and these closures had a total closure time of 9300 hours (3100 hours per year). The three years may not be representative, but the data still indicate that closures due to *impending danger* of a landslide/rock-fall/avalanche may actually have the largest share of the total cost.

Furthermore, the analysis only estimates the direct costs of delays and road repair. It does not include effects such as loss of reliability and therefore market shares for businesses, fear among the population, etc., which also will add to the total socioeconomic costs.

### 4. Summary and discussion

Key questions for the InfraRisk project were: "How vulnerable is the Norwegian transport infrastructure to extreme weather events, what values are at risk, and what measures should be undertaken to reduce the risk?". The analyses of important meteorological variables that cause problems for the roads and railroads leave little doubt that the problems we are facing

- today will persist and most likely increase in the future. Precipitation is the most important climate variable for triggering of snow avalanches and landslides of various types (debris slides, debris flows, rock falls and rock slides). Uncertainties in the analyses are mainly linked to the uncertainties in the various climate scenarios, the scarcity of observations (weather stations) and the downscaling in a complex topography. For most of the country, however, the annual precipitation has increased during the last 100 years, and will continue to increase throughout this century. Moreover, the analyses also show that the frequency
- and intensity of moderate to strong short-term precipitation events are likely to increase quite significantly. These are all trends that will lead to increased number of precipitation induced natural hazard events. Robust wind data are sparse and therefore analyses are uncertain. The trends indicate however, an increase in wind speed over Norway. Strong winds pose a problem to roads and railroads in itself, in addition to being the key factor in snow drift.
- Although relatively coarse, the risk analyses for the whole country showed the probability for a significant number of events affecting Norwegian roads and railroads every year, with large costs related to closures, detour of traffic and repair only. Other short and long term effects will add significantly to these costs. Furthermore, the potential loss of lives is not included in our analysis, and one single event may potentially cause a significant disaster. With the weather trends described above, the risk will increase further through this century. The InfraRisk project has pointed out the challenges caused by the present and future

extreme weather events, and has also provided measures to reduce the present and future risk.

A comprehensive inventory of all relevant data for the transport infrastructure is a crucial tool, which is currently lacking. This includes terrain and drainage data, weather information, snowfall data, vegetation, etc., as well as good hazard maps along the routes, in which all relevant hazards are identified, and the probability for hit can be estimated along the transport lines at all

- places where they cross hazard zones. The coarse intervals of the present hazard zones in Norway (1/5000, 1/1000, and 1/100) introduces uncertainty to the risk analysis. There is therefore an urgent need for a methodology for estimating probability for hit within the hazard zones. Modification of the present alpha beta model (Bakkehøi et al., 1983),) to estimate hit probability was initiated in the project, but more work needs to be done on refining and validation this method before it can be used along the transport lines.

Mitigation against natural hazards is expensive and is unfortunately often designed and built after an incident has occurred. In such cases the expenses increase as one has to include the costs of closure and repair in addition to the mitigation measure itself. The model for risk analysis developed in the project, although it can be further developed and improved, provides a useful tool for planners, as it may identify key problem areas and form a base for a cost-benefit analysis in order to prioritize

mitigation measures.