# Peer review of "Impacts of extreme weather events on transport infrastructure in Norway"

_Natural Hazards and Earth System Sciences, 2017_

## Short Comment (SC1) · 24 Jan 2018

**RE:** NHESS 2017 437            Frauenfelder et al. Impact of extreme weather events on transport infrastructure in Norway

**Overview**

This work shows some results about a project for the defense of road infrastructures against hydrogeological risk. The papers results a list of outputs that are not interconnected each other: susceptibility, map, rainfall threshold, runout model are listed without any link. The authors should to prepare a more organic work rather than a list of methods and results. Moreover, being NHESS a scientitifical journal, a simple list of results is not worthy of publication. I suggest the authors to stress the methodology and enhance the use of research products for practical application. In other words, this work could be valuable for scientific community in the sense of practical application of scientific approach studies.

Other issue to be stressed is the hazards this paper deals. It should be snow avalanches, rock fall and debris flow. Sometimes there are also landslide. Even if some classification assimilates debris flow to landslide, this is not really true. Debris flow is the flow of a mixture of liquid and solid with nearly the same percentages. Landslide is the movement of a solid with a small contribute in percentage of liquid. To avoid confusion, authors should better specify in the text, the considered hazards. In addition about debris flows: are they landslide-induced (Iverson, 1997) or runoff generated debris flow (Coe et al., 2008)? A specification is required.

For the next revisions please use a larger character for the text.

Moreover, the authors should introduce some definition about extreme weather event.

The writer points out the European Project PARAmount (imProved Accessibility: Reliability and security of Alpine transport infrastructure related to mountainous hazards in a changing climate) dealing about the same issues of the project whose results here shown.

The following are the detailed comments and specifications.

**Abstract**

At lines 32-33 of page 1, it seems that debris flow are not a dominant hazard as claimed at point 2.1. Authors should introduce the percentage of roads and railway threatened by debris flows.

**Introduction**

The authors should introduce some definition about extreme weather event.

**Methods**

At line 33 of page 3 what does it mean Table 1 alone? Authors should explicitly write that in Table 1 they list the main variables and, moreover, the reasons for those variables.

The statement at line 1 of page 4 ("Typically, the precipitation value………lower elevations") should be justified. Typically, in Alps the precipitation on the top is underestimated because rain gauge are usually placed at lower altitudes characterized by lower rainfall depths.

At line 18 of page 4: please write the meaning of the acronym RCM

**2.2 perhaps exposure to hazard caused by extreme weather events?**

These maps seem independent of the climate forcing, the precipitations; authors should justify their use for assessing the exposure for the increasing of the weather extreme event magnitude (i.e. the precipitation intensity).

At line 35 of page 4 it is stated that, or snow avalanche and debris flow threaten the same locations or that debris flow hazard map are built following the same procedure and means of the snow avalanche hazard maps. In the first case, this should be well explicited. In the second case, the writer has some doubts because the two phenomena are quite different and authors should justify it.

**2.3 Precipitation thresholds**

About DF (Debris Flow) I.D. the writer suggests the reading of paper of Staley et al. (2013), for its study on threshold definition, and that of Gregoretti and Dalla Fontana (2007) for the triggering rainfall definition and the comparison between different D.F. thresholds.
At line 8 of page 5, it is stated that I.D. relates the role of antecedent moisture condition of debris flow initiation. The writer has a very large doubt about it. How a rainfall event (i.e. rainfall intensity and duration) could give information on the previous rainfall to which the antecedent moisture conditions relate? This depends on the threshold. Usual D.F. thresholds use the triggering rainfall (Gregoretti and Dalla Fontana, 2007; Staley et al. 2013). If authors use another rainfall to compare to the threshold, this should be initially stated and explained. The writer gave a quick read to the work of Meyer et al. (2012). The methodology used for the thresholds is suitable for landslideinduced debris flows but not in the case of runoff-generated debris flow (Coe et al., 2008; Gregoretti and Dalla Fontana, 2008; Okano et al., 2012; Theule et al., 2012; Hurlimann et al., 2014). In runoff generated debris flow, the duration of the triggering rainfall usually range in the 15-60 minutes interval and debris flow initiation is not dependent on rainfall duration.

Moreover, in many cases, runoff generated debris flows are usually triggered when the terrain is in dry conditions where the role of the antecedent moisture conditions is negligible (this could not be the case of Norway).

Moreover, the use of the PDN day for normalizing the rainfall should be better justified. The hydrological response depends also on the terrain typology. In this case, the pdn could be not work. In addition, with climate change, the rainfalls tend to concentrated in a restricted interval so that the influence of the PDN could decreases. Then, authors should better introduce and justify the adopted thresholds.

**2,4 Risk analyses**

At line 38 of page 5, the sentence "The model aims to give information    " seems unclear.

About last sentence of page 5, the writer suggests some graphs or a brief appendix.

**3.1.4**

How the results shown in Figure 6 were obtained?

Authors just write a list without explaining the source.

**3.2.1-2**

NHESS is a scientific journal: authors should introduce something about susceptibility map and model for debris flow initiation: is that presented at 2.3? Which is the relation between rainfall threshold and orientation of slope?

Moreover, at line 30 of page 7 it should be Meyer rather than Mayer.

**3.4**

At line 22 of page 8 in landslide are also included debris flow and rockfall?

At line 8 of page 9 it should be section 3.4 rather than section 2.4

At lines 17-18 of page 9: the writer does not understand how risk modelling provides the mitigation.

Coe, J.A., Kinner D.A., Godt, J.W., 2008. Initiation conditions for debris flows generated by runoff at Chalk Cliffs, central Colorado. *Geomorphology*, 96, 270-297.

Gregoretti, C., Dalla Fontana G., 2008. The triggering of debris flow due to channel-bed failure debris flow in some alpine headwater basins of the Dolomites: analyses of critical runoff. *Hydrological Processes*. 22, 2248-2263.

Iverson R.M. The physics of debris flow. Review of Geophysics, 35, 3, 245-296.

Hurlimann M., Abanco C., Moya, J., Vilajosana I. (2014). Results and experiences gathered at the Rebaixader debris-flow monitoring site, Central Pyrenees, Spain. *Landslides*. doi:10.1007/s10346-013-0452-y 161-175

Okano K., H. Suwa, and T. Kanno (2012), Characterization of debris flows by rainstorm condition at a torrent on the Mount Yakedake volcano, Japan, *Geomorphology* 136, 88--94

Staley DM, Kean JW, Cannon SH, Schmidt KM, Laber JL. 2013. Objective definition of rainfall intensity-duration thresholds for the initiation of post-fire debris flows in southern California. *Landslides* 10: 547 – 562.

Theule, J.I., Liebault, F., Loye, A., Laigle, D., and Jaboyedoff, M., 2012. Sediment budget monitoring of debris flow and bedload transport in the Manival Torrent, SE France. *Natural Hazard Earth Sciences*, 12, 731--749

---

## Referee Comment (RC1) · W. Schwanghart (Referee) · 26 Jan 2018

W. Schwanghart (Referee)

w.schwanghart@geo.uni-potsdam.de

This manuscripts presents the work conducted during the infrarisk project, a project that investigated meteorological extreme events, avalanches, as well as mass-wasting events in the context of risk analysis of the Norwegian transport network. The paper offers an overview on the outcomes of the project and thus should take on the form of a review paper rather than a research paper. However, this distinction is not really clear from the structure and content of the manuscript. Rather, the paper follows the outline of a typical research paper (Introduction, Methods, Results, Discussion and summary, Conclusions) and thus suffers from being a rather hybrid form between a research and review paper without meeting the demands of any of the two.

[Figure]

As a research paper, the manuscript presents insufficient novelty and originality as it largely iterates the research conducted during the project. As a review paper, it fails to cover the breadth and scope of the topic. Although I do not know the project very well, I know that some work - that was part of the project - has not been referenced although some of it is directly related to the topic of the paper (see Meyer et al. 2015 (of which I was a coauthor)). I am not demanding to be cited, but I find it too minimalistic for a review to base some of the results on mainly one research paper (Dyrrdal et al. 2012).

I thus recommend to abandon the research-paper structure. The paper suffers anyway from not adhering to this formal structure. Many statements in the Results should better be placed in a Discussion (page 7, line 20f) or the Methods (page 7, line 27f).

A major benefit and novelty of this paper could be to detail how the results obtained in the project are now being used in policy and decision making. Rather than reading repetitive truisms such as "complete protection against natural hazards [...] is not possible", readers might want to know how the project's result have impacted policies towards the management of infrastructure-related risks. Since the project was finalized in 2013, such information may be readily available. As it stands, the manuscripts largely summarizes previously published results, and a clear added value is hard to discern.

I do not recommend to reject the paper, because I think that it potentially provides an insightful report of a large project on a timely and relevant issue in natural hazards and risk research. However, the manuscript requires substantial work and thus I recommend major revisions.

References

Meyer, N. K., Schwanghart, W., Korup, O. and Nadim, F.: Roads at risk: traffic detours from debris flows in southern Norway, Nat. Hazards Earth Syst. Sci., 15(5), 985–995, doi:10.5194/nhess-15-985-2015, 2015.

---

## Editor Comment (EC1) · P. Tarolli (Editor) · 15 Mar 2018

I think that the authors should provide at this stage their reply to reviewer's comments. The short comment SC1 was uploaded wrongly as "short comment" instead of "referee comment".